# Implicit Bias in Matrix Factorization and its Explicit Realization in a New Architecture

## Abstract

Gradient descent for matrix factorization is known to exhibit an implicit bias toward approximately low-rank solutions. While existing theories often assume the boundedness of iterates, empirically the bias persists even with unbounded sequences. We thus hypothesize that implicit bias is driven by divergent dynamics markedly different from the convergent dynamics for data fitting. Using this perspective, we introduce a new factorization model: $X \approx U D V^\top$, where $U$ and $V$ are constrained within norm balls, while $D$ is a diagonal factor allowing the model to span the entire search space. Our experiments reveal that this model exhibits a strong implicit bias regardless of initialization and step size, yielding truly (rather than approximately) low-rank solutions. Furthermore, drawing parallels between matrix factorization and neural networks, we propose a novel neural network model featuring constrained layers and diagonal components. This model achieves strong performance across various regression and classification tasks while finding low-rank solutions, resulting in efficient and lightweight networks.

## 1 Introduction

The Burer–Monteiro (BM) factorization (Burer & Monteiro, 2003) is a classical technique for obtaining low-rank solutions in optimization. One can view it as a simple neural network that uses a single layer of hidden neurons under linear activation. Indeed, given the factorization $X = UV^T$ where $U \in \mathbb{R}^{d \times r}$ and $V \in \mathbb{R}^{c \times r}$, one can view $U$ and $V$ as the weights of the first and second layers, and $r$ as the number of hidden neurons. But despite the similarity suggested by this view, there is a clear distinction between BM factorization and neural networks in how the rank $r$ is chosen. In BM, $r$ is typically chosen to be small, close to the rank of the desired solution. Neural networks, on the other hand, often succeed even in overparametrized settings where $r$ is large.

Recent findings of implicit regularization in matrix factorization narrow the gap between these two perspectives. For instance, Gunasekar et al. (2017) demonstrate that gradient descent (with certain parameter selection) on BM factorization tends to converge toward approximately low-rank solutions even when $r = d$. Based on this observation, they conjecture that *"with small enough step sizes and initialization close enough to the origin, gradient descent on full-dimensional factorization converges to the minimum nuclear norm solution."*

In a follow-up work, Razin & Cohen (2020) present a counterexample demonstrating that implicit regularization in BM factorization *cannot be explained by minimal nuclear norm*, or in fact any norm. Specifically, they show that there are instances where the gradient method applied to BM factorization yields a diverging sequence, and all norms thus grow toward infinity. Intriguingly, despite this divergence, they found that the rank of the estimate decreases toward its minimum.

Although this phenomenon might seem surprising initially, it is not uncommon for diverging sequences to follow a structured path. A prime example is the Power Method, the fundamental algorithm

for finding the largest eigenvalue and eigenvector pair of a matrix. Starting from a random initial point $x_0$, the Power Method iteratively updates the estimate by multiplying it with the matrix. This process amplifies the component of the vector that aligns with the direction of the dominant eigenvector more than the other components, progressively leading $x_k$ to align with this eigenvector. In practical implementations, $x_k$ is scaled after each iteration to avoid numerical issues from divergence.

This perspective underpins our approach. Specifically, our key insight is that the implicit regularization in BM factorization (and neural networks) is driven by divergent dynamical behavior. This is markedly different from the standard (convergent) optimization dynamics helping with the data fitting. In this context, we hypothesize that these forces do not merely coexist but actively compete, influencing model behavior and performance in fundamentally conflicting ways. Our main goal in the development of this paper is to devise an approach that unravels these competing forces.

## 1.1 Overview of main contributions

- **A novel formulation for matrix factorization**. We model $X = UDV^\top$, where $U$ and $V$ are constrained within Frobenius norm balls. Projection onto this ball results in a scaling step similar to the Power Method. The middle term $D$ is a diagonal matrix that allows the model to explore the entire search space despite $U$ and $V$ being bounded.[1]

  Through extensive empirics we demonstrate that the gradient method applied to the proposed formulation exhibits a pronounced implicit bias toward low-rank solutions. We compare our formulation against standard BM factorization with two unconstrained factors. Specifically, we investigate key factors such as step size and initialization, which prior work suggests might be contributing to implicit bias. We find that our factorization approach largely obviates the need to rely on these conditions: it consistently finds *truly (rather than approximately) low-rank solutions* across a wide range of initializations and step-sizes in our experiments. We believe these findings should be of broader interest to research on implicit bias.

- **A novel architecture**. Motivated by the strong bias for low-rank solutions of the proposed factorization, we subsequently extend it to deep neural networks. We do so by adding constrained layers and diagonal components. We show that this constrained model performs on par with, or even better than, the standard architecture across various regression and classification tasks. Importantly, our approach exhibits bias towards low-rank solutions, resulting in a natural pruning procedure that delivers compact, lightweight networks without compromising performance.

## 1.2 Related Work

**Burer-Monteiro factorization.** BM factorization was proposed for solving semidefinite programs (Burer & Monteiro, 2003, 2005) and has been recognized for its efficiency in addressing low-rank optimization problems (Boumal et al., 2016; Park et al., 2018). Building on the connections between matrix factorization and training problems for two-layer neural networks, BM models have served as foundational building blocks for understanding implicit bias and developing theoretical insights.

**Implicit regularization.** One promising line of research that aims to explain the successful generalization abilities of neural networks is that of 'implicit regularization' induced by the optimization methods and architectures (Neyshabur et al., 2014, 2017; Neyshabur, 2017). Several studies explore matrix factorization to investigate implicit bias (Gunasekar et al., 2017; Arora et al., 2018; Razin & Cohen, 2020; Belabbas, 2020; Li et al., 2021). Much of the existing work focuses on gradient flow dynamics in the limit of infinitesimal learning rates. Exceptionally, Gidel et al. (2019) examine discrete gradient dynamics in two-layer linear neural networks, showing that the dynamics progressively learn solutions of reduced-rank regression with a gradually increasing rank.

**Constrained neural networks.** Regularizers are frequently used in neural network training to prevent overfitting and improve generalization, or to achieve structural benefits such as sparse and compact network architectures (Scardapane et al., 2017). However, it is conventional to apply these regularizers as penalty functions in the objective rather than constraints. This approach is likely favored due to the ease of implementation, as pre-built functions are readily available in common neural network packages. Regularization in the form of constraints appears to be rare in neural network training. One notable exception is in the context of neural network training with the Frank-Wolfe algorithm (Pokutta et al., 2020; Zimmer et al., 2022; Macdonald et al., 2022). Recently, Pethick et al. (2025)

---

[1]The reader may notice a "syntactic" similarity with SVD; except using vastly simpler Frobenius norm constraints on $U$ and $V$ instead of orthogonality.

revealed parallels between Frank-Wolfe on constrained networks and algorithms that post-process update steps, such as Muon (Jordan et al., 2024), which achieves state-of-the-art results on nanoGPT by orthogonalizing the update directions before applying them.

**Pruning.** Neural networks are overparameterized, which can enhance generalization and avoid poor local minima. But such models then suffer from excessive memory and computational demands, making them less efficient for deployment in real-world applications (Chang et al., 2021). Pruning reduces the number of parameters, resulting in more compact and efficient models that are easier to deploy. A comprehensive review on pruning is beyond the scope of this paper due to space limitations and the diversity of approaches. We refer to (Reed, 1993; Blalock et al., 2020; Cheng et al., 2024) and the references therein for detailed reviews. Pruning by singular value thresholding has recently shown promising results, particularly in natural language processing (Chen et al., 2021), and is often used along with various enhancements such as importance weights and data whitening for effective compression of large language models (Hsu et al., 2022; Yuan et al., 2023; Wang et al., 2024).

## 2 Matrix Factorization with a Diagonal Component

Consider *matrix sensing*, a problem where we seek to recover a *positive semidefinite* (PSD) matrix $X \in \mathbb{S}_+^{d \times d}$ from a set of linear measurements $b = \mathcal{A}(X) \in \mathbb{R}^n$. We define $\mathcal{A} : \mathbb{R}^{d \times d} \to \mathbb{R}^n$ through symmetric measurement matrices $A_1, \ldots, A_n \in \mathbb{S}^{d \times d}$, such that $\mathcal{A}(X) = [\langle A_1, X \rangle \ \cdots \ \langle A_n, X \rangle]^\top$ and $\mathcal{A}^\top y = \sum_{i=1}^n y_i A_i$. We particularly focus on the data-scarce setting where $n \ll d^2$. A notable example here matrix completion, where one completes a matrix $X$ given a subset of its entries. This problem is inherently under-determined; but successful recovery is possible if $X$ is low-rank (Candes & Recht, 2012). We focus on recovering a PSD matrix for simplicity; this is without loss of generality, as the general case can be be easily reformulated as a PSD matrix sensing problem (Park et al., 2017).

The problem described above can be cast as the following rank-constrained optimization problem:

$$\min_{X \in \mathbb{S}_+^{d \times d}} \quad f(X) := \tfrac{1}{2}\|\mathcal{A}(X) - b\|_2^2 \quad \text{subj. to} \quad \text{rank}(X) \le r. \tag{1}$$

Although rank-constrained matrix optimization problems are typically NP-hard, various methods have been developed to provide practical approximations. One prominent approach is BM factorization, which reparametrizes the decision variable $X$ as $UU^\top$, where the factor $U \in \mathbb{R}^{d \times r}$, and $r$ is a positive integer that controls the rank of the resulting product. Problem (1) can then be reformulated as:

$$\min_{U \in \mathbb{R}^{d \times r}} \quad \tfrac{1}{2}\|\mathcal{A}(UU^\top) - b\|_2^2. \tag{2}$$

Despite the fact that finding the global minimum of (2) remains challenging, a local solution can be approximated using gradient descent (Lee et al., 2016). Initializing at $U_0 \in \mathbb{R}^{d \times r}$, perform:

$$U_{k+1} = U_k - \eta \nabla_U f(U_k U_k^\top), \tag{3}$$

where $\eta > 0$ is the step-size, and the gradient is computed as $\nabla_U f(UU^\top) = 2\nabla f(UU^\top)U$.

Selecting the factorization rank $r$ is a critical decision. A small $r$ may lead to spurious local minima, resulting in inaccurate outcomes (Waldspurger & Waters, 2020). Conversely, a large $r$ might weaken rank regularization, rendering the problem underdetermined. Conventional wisdom in BM factorization suggests finding a moderate compromise between these two extremes. However, a key observation in (Gunasekar et al., 2017) is that the gradient method applied to (2) exhibits a tendency towards approximately low-rank solutions even when $r = d$. Below, we restate their conjecture:

**Conjecture in (Gunasekar et al., 2017).** Suppose gradient flow (*i.e.,* gradient descent with an infinitesimally small step-size) is initialized at a *full-rank matrix arbitrarily close to the origin*. If the limit of the gradient flow, $X_{\text{GF}} = UU^\top$, exists and is a global optimum of (1) with $\mathcal{A}(X_{\text{GF}}) = b$, then $X_{\text{GF}}$ is the minimal nuclear-norm solution to (1).

### 2.1 The Proposed Factorization

We propose reparameterizing $X = UDU^\top$, where $U \in \mathbb{R}^{d \times r}$ is constrained to have a bounded norm, and $D \in \mathbb{R}^{r \times r}$ is a non-negative diagonal matrix:

$$\min_{\substack{U \in \mathbb{R}^{d \times r} \\ D \in \mathbb{R}^{r \times r}}} \quad \frac{1}{2}\|\mathcal{A}(UDU^\top) - b\|_2^2 \quad \text{s.t.} \quad \|U\|_F \le \alpha, \ \ D_{ii} \ge 0, \ \ D_{ij} = 0, \ \ \forall i \text{ and } \forall j \ne i, \tag{4}$$

where $\alpha > 0$ is a model parameter. When the problem is well-scaled, for instance through basic preprocessing with data normalization, we found that $\alpha = 1$ is a reasonable choice.

Placing in multiple factors and with constraints, we perform projected-gradient updates on $U$ and $D$ with step-size $\eta > 0$:

$$
\begin{aligned}
U_{k+1} &= \Pi_U \left( U_k - \eta \nabla_U f(U_k D_k U_k^\top) \right) \\
D_{k+1} &= \Pi_D \left( D_k - \eta \nabla_D f(U_k D_k U_k^\top) \right),
\end{aligned}
\tag{5}
$$

where $\Pi_U$ and $\Pi_D$ are projections for the constraints in (4); while the gradients are

$$
\nabla_U f(UDU^\top) = 2\nabla f(UDU^\top)UD \quad \text{and} \quad \nabla_D f(UDU^\top) = U^\top \nabla f(UDU^\top)U.
$$

## 2.2 Numerical Experiments on Matrix Factorization

We present numerical experiments comparing the empirical performance of the proposed approach with the classical BM factorization. Specifically, we examine the impact of initialization and step-size on the singular value spectrum of the resulting solution. We set up a synthetic matrix completion problem to recover a PSD matrix $X_\natural = U_\natural U_\natural^\top$, where the entries of $U_\natural \in \mathbb{R}^{100 \times 3}$ are drawn independently from $N(0,1)$. We randomly sample $n = 900$ entries of $X_\natural$ and store them in the vector $b \in \mathbb{R}^n$. The goal is to recover $X_\natural$ from $b$ by solving problems (2) and (4). For initialization, we generate $U_0 \in \mathbb{R}^{d \times d}$ with entries drawn independently from $N(0,1)$; we rescale $U_0$ to have Frobenius norm $\xi > 0$ (we investigate the impact of $\xi$). We initialize $D_0 = I$.

The results are shown in Figure 1. First, we examine the impact of step-size. To this end, we fix $\xi = 10^{-2}$ and test different values of $\eta$. In the left panel, we plot the objective residual as a function of iterations. As expected, we observe that a smaller step-size slows down convergence. In the right panel, we plot the singular value spectrum of the results attained after $10^6$ iterations. We observe no direct connection between step-size and implicit bias in BM factorization.

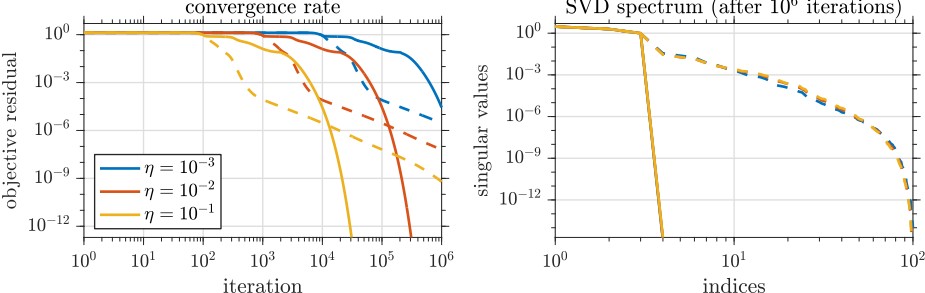

Impact of **step-size** ($\eta$), in **noiseless** setting, with fixed initialization.

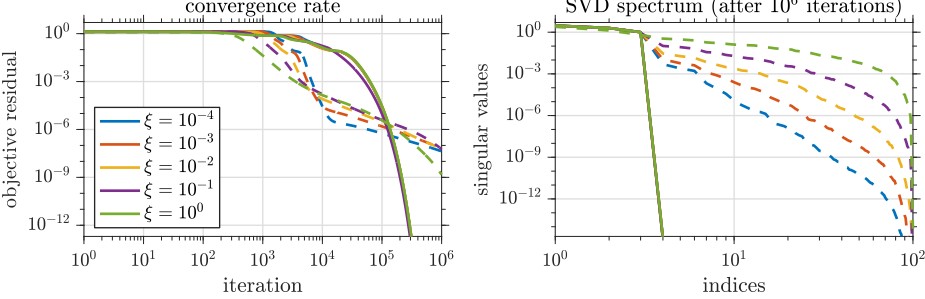

Impact of **initial distance to origin** ($\xi$), in **noiseless** setting, with fixed step-size.

Figure 1: Impact of step-size and initialization on implicit bias. **Solid lines represent our UDU factorization**, while **dashed lines denote the classical BM factorization**. [*Left*] Objective residual vs. iterations. [*Right*] Singular value spectrum after $10^6$ iterations. In all cases, UDU produces truly low-rank solutions, whereas the classical approach results in approximate low-rank structures.

Next, we investigate the impact of initialization. We fix the step-size at $\eta = 10^{-2}$ and evaluate the effect of varying $\xi$. We observe a correlation between the implicit bias of the BM factorization and $\xi$,

which determines the initial distance from the origin. Initializing closer to the origin in the classical BM factorization yields solutions with a faster spectral decay. Notably, the UDU factorization demonstrates a strong implicit bias toward truly low-rank solutions, regardless of the choice of $\eta$ or $\xi$.

We provide additional experiments in the Appendices. Specifically, Appendix A.1 considers the matrix completion problem with noisy measurements. The results remain consistent with the noiseless case: the UDU model exhibits an implicit bias toward truly low-rank solutions, while the classical BM factorization yields approximately low-rank solutions. Additionally, we present numerical experiments on a matrix sensing problem arising in phase retrieval image recovery in Appendix A.2. As before, the UDU framework consistently promotes low-rank solutions, and this structural bias significantly enhances the quality of the recovered image.

## 2.3 Theoretical Insights into the Inner Workings and Implicit Bias

A fixed-point analysis of the proposed method provides valuable insights into its inner workings. Define the update variables before projection as $\bar{U} = U - 2\eta\nabla f(X)UD$ and $\bar{D} = D - \eta U^\top \nabla f(X)U$, with $X = UDU^\top$. Suppose $(U, D)$ is a fixed point of the algorithm in (5). Then, the following hold:

Let $u_j$ denote the $j^{\text{th}}$ column of $U$ and $\lambda_j$ the $j^{\text{th}}$ diagonal entry of $D$.

$(a)$ If $\|\bar{U}\| \leq \alpha$, then $\nabla f(X)u_j\lambda_j = 0$ for all $j$,

$(b)$ If $\|\bar{U}\| > \alpha$, then there exists some $\beta > 0$ such that $\nabla f(X)u_j\lambda_j = -\beta u_j$ for all $j$.

At this point, it may seem that choosing a small value of $\alpha$ could promote a fixed point where the columns of $U$ align with the negative eigenvectors of $\nabla f(X)$. However, as we will see from the analysis of $D$, there are no valid fixed points that satisfy $\|\bar{U}\| > \alpha$, since

$(c)$ If $\lambda_j = 0$, then $u_j^\top\nabla f(X)u_j \geq 0$,  , while   $(d)$ If $\lambda_j > 0$, then $u_j^\top\nabla f(X)u_j = 0$.

Suppose $\|\bar{U}\| > \alpha$. Then, $(b)$ implies that if $\lambda_j > 0$, then $u_j$ must be an eigenvector of $\nabla f(X)$ corresponding to a negative eigenvalue; and if $\lambda_j = 0$, then $u_j$ must also be zero. However, the first statement contradicts $(d)$, while the second statement agrees with $(c)$ only if $u_j = 0$. Since these conditions must hold for all $j$, it follows that $U = 0$. This, in turn, implies that $\bar{U} = 0$, which contradicts the initial assumption that $\|\bar{U}\| > \alpha$, hence there are no fixed points satisfying $\|\bar{U}\| > \alpha$.

Considering $(a)$, observe that the fixed point characterization obtained here coincides with the fixed points of the BM factorization after the change of variables from $U$ to $UD^{1/2}$. Thus, incorporating constraints on $U$ and adding the factor $D$ does not introduce any new fixed points for $X$.

Interestingly, this fixed point analysis also provides insight into the low-rank bias of the algorithm. In particular, when $U$ tends to grow and $\|\bar{U}\|$ exceeds $\alpha$, the algorithm appears to temporarily favor directions where the columns of $U$ align with the negative eigenvectors of $\nabla f(X)$. To see this more concretely, we can express the update rules in terms of the columns of $\bar{U}$ and the diagonal entries of $\bar{D}$ as $\bar{u}_j = u_j - 2\eta\nabla f(X)u_j\lambda_j$ and $\bar{\lambda}_j = \lambda_j - \eta u_j^\top\nabla f(X)u_j$. These expressions show that both $\bar{u}_j$ and $\bar{\lambda}_j$ tend to grow in the directions aligned with the negative eigenvectors of $\nabla f(X)$. However, from statement $(d)$, we know that there is no fixed point with $\lambda_j > 0$ unless $u_j^\top\nabla f(X)u_j = 0$. This suggests that: (i) either the algorithm might push $u_j$ towards zero, which could happen only through the projection steps if another column $\bar{u}_{j'}$ exhibits a faster growth; or (ii) $X$ should evolve such that $f(X)$ is minimized in the direction of $u_j$, effectively moving towards a point where $\nabla f(X)u_j = 0$.

The analysis presented here is a simplified perspective aimed at gaining insight. In reality, the alignment or shrinking of the columns of $U$ and the minimization of $f(X)$ along specific directions reflected in these columns occur simultaneously and interact in a complex manner. Nevertheless, we can clearly observe these effects in our numerical experiments. In Appendix A.3, we present the evolution of the column norms of $U$ and the diagonal entries of $D$ over the iterations in our matrix completion experiment. Our results show that initially, a few specific columns of $U$ grow, pushing all other columns numerically to zero. Once $f(X)$ is effectively minimized with respect to these initial columns, some other columns are identified and start to grow. Eventually, the algorithm converges to a low-rank solution, where the factorization $UDU^\top$ is rank-revealing since only a few columns of $U$ are nonzero. We further observe that these nonzero columns are orthogonal, effectively demonstrating how the algorithm's specific preference to align $u_j$ with the negative eigenvectors of $\nabla f(X)$ along the path implicitly induces a structured solution.

 ## 3 Feedforward Neural Networks with Diagonal Hidden Layers

This section extends our approach to neural networks. Consider a dataset comprising $n$ data points $(\mathbf{x}_i, \mathbf{y}_i) \in \mathbb{R}^d \times \mathbb{R}^c$. We first define a three-layer neural network defined as

$$\phi(\mathbf{x}) := \sum_{j=1}^{m} \mathbf{v}_j w_j \mathbf{u}_j^\top \mathbf{x} \approx \mathbf{y}. \tag{6}$$

The first and third layers are fully connected, and the middle is a diagonal layer, as illustrated in Figure 2. Drawing parallels between our matrix factorization model in (4) and neural network training, we impose Euclidean norm constraints on the weights of the fully connected layers. Under these conditions, the training problem can be formulated as follows:

$$\min_{\mathbf{u}_j, w_j, \mathbf{v}_j} \quad \frac{1}{2n} \sum_{i=1}^{n} \| \sum_{j=1}^{m} \mathbf{v}_j \, w_j \, \mathbf{u}_j^\top \mathbf{x}_i - \mathbf{y}_i \|_2^2$$
$$\text{subj.to} \quad \sum_{j=1}^{m} \|\mathbf{u}_j\|_2^2 \leq 1, \quad \sum_{j=1}^{m} \|\mathbf{v}_j\|_2^2 \leq 1, \text{ and } w_j \geq 0; \quad \text{for all } j = 1, \ldots, m. \tag{7}$$

The norm constraints in our training problem can be interpreted as a stronger form of weight decay, one of the most commonly used regularization techniques in neural networks, which lends further justification to our formulation. We refer to this neural network structure as UDV.

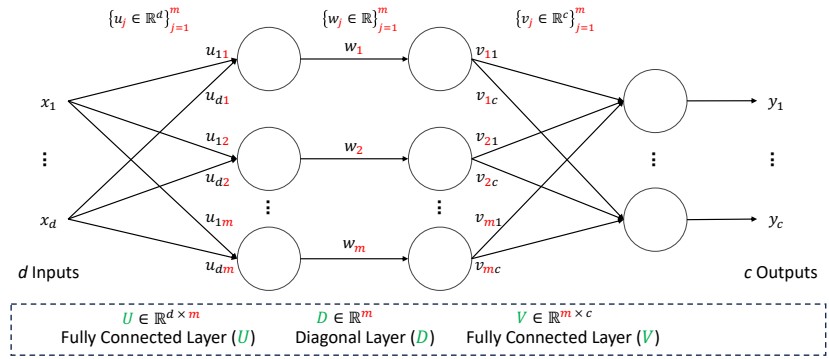

Figure 2: UDV structure. The weights in diagonal layer $D$ are denoted as $w_j$.

### 3.1 Numerical Experiments on Neural Networks

In this section, we test the proposed UDV framework on regression and classification tasks, comparing it with fully connected two-layer neural networks (denoted as UV in the subsequent text) using both linear and ReLU activation functions. This comparison is fair in terms of computational cost, as the cost incurred by the diagonal layer —which can also be viewed as a parameterized linear activation function— is negligible. We observe a strong empirical bias toward low-rank solutions in all our experiments. We also present a proof-of-concept use case of this strong bias, combined with an SVD-based pruning strategy, to produce compact networks.

#### 3.1.1 Implementation Details

**Computing environment.** All classification tasks were conducted on an NVIDIA A100 GPU with four cores of the AMD Epyc 7742 processor, while regression tasks were conducted on a single core of an Intel Xeon Gold 6132 processor. We used Python 3.9.5 and PyTorch 2.0.1.

**Datasets.** We used two datasets for the regression tasks: House Prices - Advanced Regression Techniques (HPART) (Anna Montoya, 2016) and New York City Taxi Trip Duration (NYCTTD) (Risdal, 2017). We allocated 80% of the data for the training and reserved the remaining 20% for validation. Following (Huang, 2003), we set the number of hidden neurons in the diagonal layer $m = \texttt{round}\big(\sqrt{(c+2)d} + 2\sqrt{d/(c+2)}\big)$. This results in a network structure ($d$-$m$-$c$) of 79-26-1 for HPART, and 12-10-1 for NYCTTD.

For classification tasks, we used the normalized MNIST dataset (LeCun et al., 2010). We applied transfer learning by replacing the classifier layers of three advanced neural networks –MaxViT-T (Tu

Table 1: Model performance using different models. M, E, and R represent the transferred models MaxVit-T, EfficientNet-B0, and RegNetX-32GF, respectively.

| Tasks | Regression (Test Loss) | | Classification (Test Accuracy) | | |
|---|---|---|---|---|---|
| Dataset | HPART | NYCTTD | MNIST | | |
| UDV | $1.304 \times 10^{-3}$ Adam: $10^{-3}$ | $5.248 \times 10^{-6}$ NAdam: $10^{-4}$ | M: 99.67% MBGDM: $10^{-2}$ | E: 99.63% MBGDM: $10^{-1}$ | R: 99.74% MBGDM: $10^{-2}$ |
| UV | $1.333 \times 10^{-3}$ Adam: $10^{-3}$ | $5.251 \times 10^{-6}$ Adam: $10^{-3}$ | M: 99.69% MBGDM: $10^{-2}$ | E: 99.60% Adam: $10^{-3}$ | R: 99.66% MBGD: $10^{0}$ |
| UV-ReLU | $1.167 \times 10^{-3}$ Adam: $10^{-3}$ | $5.323 \times 10^{-6}$ NAdam: $10^{-3}$ | M: 99.68% NAdam: $10^{-4}$ | E: 99.68% MBGDM: $10^{-1}$ | R: 99.73% MBGD: $10^{0}$ |

et al., 2022), EfficientNet-B0 (Tan & Le, 2019), and RegNetX-32GF (Radosavovic et al., 2020)– with UDV, while using pre-trained weights from ImageNet-1K (Deng et al., 2009). Specifically, we retained all layers up to the first fully connected layer of the classifier and replaced the subsequent layers. The number of hidden neurons in the diagonal layer was set to as $m = \texttt{floor}(\frac{2}{3}d)$. This results in a UDV network structure ($d$-$m$-$c$) of 512-341-10 for MaxViT-T, 1280-853-10 for EfficientNet-B0, and 2520-1680-10 for RegNetX-32GF.

**Loss function.** We used mean squared error for regression and cross-entropy loss for classification.

**Optimization methods.** We tested the results using four different optimization algorithms for training: Adam (Kingma & Ba, 2014), Mini-Batch Gradient Descent (MBGD) (LeCun et al., 2002), NAdam (Dozat, 2016), and Mini-Batch Gradient Descent with Momentum (MBGDM) (Sutskever et al., 2013). For classification, we used different batch sizes for different models: 128 for MaxViT-T and RegNetX-32GF, and 384 for EfficientNet-B0.

We tuned the step size for all models and optimization algorithms: For regression tasks, we tested step sizes $10^{-4}$, $10^{-3}$, $10^{-2}$, $10^{-1}$, 1, 2, 3. Larger step sizes (1, 2, 3) were often excluded for the $UV$ model due to divergence. For classification we tested LRs $10^{-6}$, $10^{-5}$, $10^{-4}$, $10^{-3}$, $10^{-2}$, $10^{-1}$, 1 with Adam and NAdam; and we tested $10^{-3}$, $10^{-2}$, $10^{-1}$, 1, 2, 3, 5 with MBGD and MBGDM.

**Training Procedure.** UV and UDV models were initialized identically, with the diagonal elements of $D$ initialized using Kaiming Uniform Initialization, consistent with the default initialization for fully connected layers in PyTorch. The models were trained for 200 epochs on HPART, 50 epochs on NYCTTD, and 70 epochs on MNIST; and results were averaged over 1000 random seeds for HPART, 100 for NYCTTD, and 1 for MNIST to ensure robustness. The validation loss, used as a generalization metric in regression, was averaged over the final 20 epochs for the HPART dataset and the final 5 epochs for the NYCTTD dataset. Similarly, validation accuracy for classification tasks was averaged over the last 5 epochs to ensure stability in the reported values.

### 3.1.2 Low-rank Bias in Neural Network Training

Table 1 presents the validation loss (for regression) or validation accuracy (for classification) of the UDV model compared to the classical UV model with linear and ReLU activation functions. For each configuration (dataset and model architecture), the results are obtained by selecting the best algorithm and learning rate pair. Moreover, Figure 3 illustrates the singular value spectrum of the solutions corresponding to each entry in these tables. We focus on the singular values from the $U$ and $UD$ layers, as they generate the primary data representation, while omitting the $V$ layer, which serves as the feature selection layer and is a tall matrix by definition, given that $c \ll m$ in most cases. Collectively, these results show that the UDV framework achieves competitive prediction accuracy while exhibiting a strong implicit bias toward low-rank solutions, as indicated by the faster decay in the singular value spectrum.

### 3.1.3 Reducing Network Size with SVD-based Pruning

Efficient and lightweight feed-forward layers are crucial for real-world applications. For instance, the Apple Intelligence Foundation Models (Gunter et al., 2024) recently reported that pruning hidden dimensions in feed-forward layers yields the most significant gains in their foundation models.

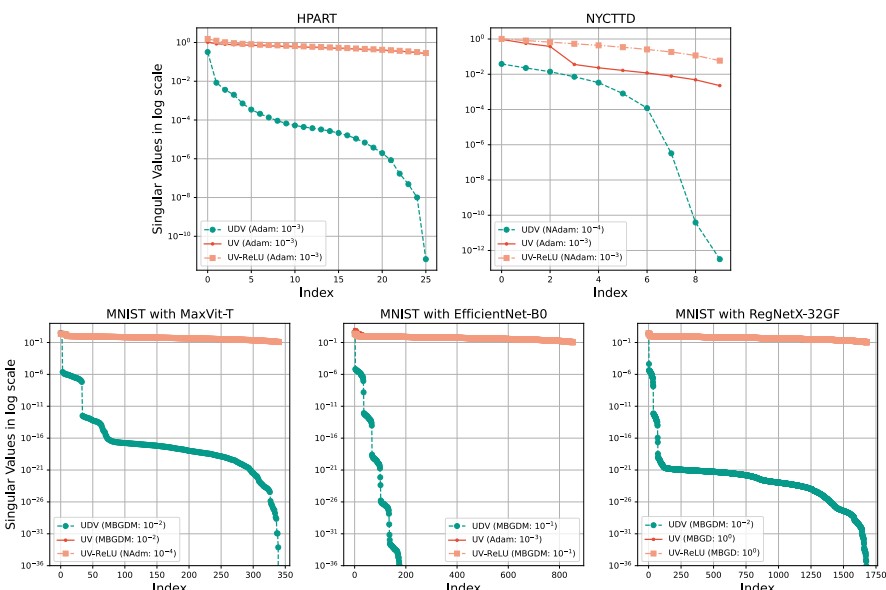

Figure 3: Singular value spectrum corresponding to the solutions reported in Table 1.

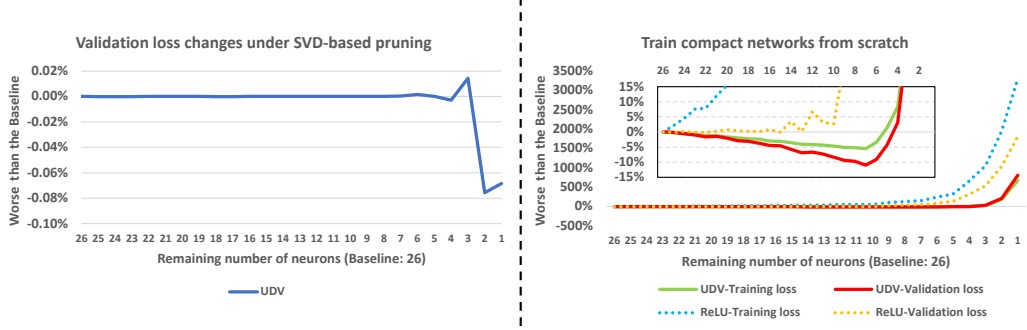

Figure 4: Comparison of SVD-based pruning and re-training compact networks on the HPART dataset using the NAdam algorithm with a learning rate of $10^{-3}$. Negative percentages indicate improvements over the baseline. SVD-based pruning demonstrates that the UDV leads to a compact model without performance degradation, while retraining shows that the UDV achieves better generalization in a compact model.

Building on this insight, we leverage the inherent low-rank bias of the UDV architecture through an SVD-based pruning strategy to produce compact networks without sacrificing performance.

A low-rank solution was observed when applying SVD to $UD$ layers:

$$UD = \mathtt{USV}^\top, \quad \mathtt{U} \in \mathbb{R}^{d \times m}, \quad \mathtt{S} \in \mathbb{R}^{m \times m}, \quad \mathtt{V}^\top \in \mathbb{R}^{m \times m}. \tag{8}$$

By dropping small singular values in $\mathtt{S}$, these matrices can be truncated to $\bar{\mathtt{U}} \in \mathbb{R}^{d \times r}, \bar{\mathtt{S}} \in \mathbb{R}^{r \times r}$ and $\bar{\mathtt{V}}^\top \in \mathbb{R}^{r \times m}$, where $0 < r < m$. Consequently, $(m - r)$ neurons can be pruned, and new weight matrices are assigned:

$$\bar{U} = \bar{\mathtt{U}} \in \mathbb{R}^{d \times r}, \quad \bar{D} = \bar{\mathtt{S}} \in \mathbb{R}^{r \times r}, \quad \bar{V} = \bar{\mathtt{V}}^T V \in \mathbb{R}^{r \times c}. \tag{9}$$

We applied this pruning strategy on the models from Table 1. The left part of Figure 4 presents an example comparing the generalization capability of pruned models. For comparison, we created compact models by training from scratch with a reduced number of neurons $m$ in the hidden layer. The performance change for these models is shown in the right panel of Figure 4. Although our pruned networks derived from UDV demonstrate that models with significantly fewer parameters can still achieve strong generalization, these compressed architectures are often more challenging to optimize directly within the reduced space, consistent with prior findings in the literature (Arora et al., 2018; Chang et al., 2021). We omit the results for retraining with the UV model, as they show similar trends to UDV in this context, though UDV generally exhibits superior generalization.

### 3.1.4 Further Details and Discussions

Our findings in experiments on neural networks align with the results observed in matrix factorization. A key distinction, however, was the use of different optimization algorithms, including stochastic gradients and momentum steps, in neural network experiments. Despite these differences, the UDV architecture consistently demonstrated a strong bias toward low-rank solutions. Additional experiments and details can be found in the Appendices, with key results summarized below:

- In the early stages of this research, we explored four variants of UDV, each differing slightly in their constraints. We selected the version presented in (7), as it generally exhibits the most pronounced decay in the singular value spectrum. For completeness, details of the other three variants are provided in Appendix B.1.
- Appendix B.2 provides additional results for the experiment described in Section 3.1.2. Specifically, we present results analogous to those in Table 1 and Figure 3, but focusing exclusively on the MBGDM algorithm. These results exhibit similar trends, reinforcing consistency across different methods. Additionally, comprehensive performance comparisons across all algorithms and models are provided in Tables SM2 to SM5 in the Appendices.
- Additional results on SVD-based pruning are provided in Appendix B.3. We demonstrate that the UDV framework consistently achieves low-rank solutions across various problem configurations. Furthermore, we analyze the effect of learning rate on the singular value spectrum, similar to the analysis in Figure 1, but applied to neural network experiments. This analysis confirms that the UDV framework produces low-rank solutions across a broad range of learning rates.
- Appendix B.4 extends the UDV framework by incorporating ReLU activation. Preliminary experiments with the UDV-ReLU model indicate that it also tends to yield low-rank solutions, similar to those observed in the original UDV framework.
- Prior work on implicit bias in neural networks suggests that increasing depth enhances the tendency toward low-rank solutions (Arora et al., 2019; Feng et al., 2022), raising the question of whether the pronounced bias in the UDV framework is just a consequence of adding a diagonal layer. To investigate this, we conducted experiments comparing the UDV model to fully connected three-layer networks, as detailed in Appendix B.5. Additionally, we included a UDV model without constraints in these comparisons. The results indicate that this bias cannot be attributed solely to depth, highlighting the critical role of explicit constraints.
- Appendix B.6 compares the spectral decay of the UDV network to that induced by classical weight decay regularization in two- and three-layer networks. While weight decay promotes singular value decay, it can not reproduce the strong decay observed in the UDV model.
- Appendix B.7 presents a toy example demonstrating the application of the UDV structure within the LoRA framework to fine-tune a pre-trained LLaMA-2 model on a causal language modeling task. The pruning results highlight the potential of the proposed UDV block to replace linear layers in a wide range of models, demonstrating promising performance under compression.

## 4  Conclusions

We proposed a new matrix factorization framework, inspired by the observation that implicit bias is driven by dynamics that are distinct from those leading to convergence of the objective function. This framework constrains the factors within Euclidean norm balls and introduces a middle diagonal factor to ensure the search space is not restricted. Numerical experiments demonstrate that this approach significantly strengthens the low-rank bias in the solution.

To explore the broader applicability of our findings, we designed an analogous neural network architecture with three layers, constraining the fully connected layers and adding a diagonal hidden layer, referred to as UDV. Extensive experiments show that the proposed UDV architecture achieves competitive performance compared to standard fully connected networks, while inducing a structured solution with a strong bias toward low-rank representations. Additionally, we explored the utility of this low-rank structure by applying an SVD-based pruning strategy, illustrating how it can be leveraged to construct compact networks that are more efficient for downstream tasks.

The proposed model exhibits reduced rank regression behavior, where the training process gradually increases the model rank. While this promotes a low-rank structure, it can also slow convergence. Although we provide some theoretical insights, developing a more complete theory and designing algorithms that fully exploit the model's regularization capabilities, especially for large-scale problems, remain important directions for future work.

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
