# OpenReview forum: "Implicit Bias in Matrix Factorization and its Explicit Realization in a New Architecture"
_NeurIPS.cc/2025/Conference — Submitted to NeurIPS 2025_

### Official Review · Reviewer_g4tS · 2025-06-06

**Clarity:** 3
**Significance:** 2
**Originality:** 3
**Rating:** 4
**Confidence:** 4

**Summary:**

Introduces a new UDV matrix factorization model $X = UDV^\top$ (and a symmetric $X = UDU^\top$ variant), inspired by the implicit bias towards low rank in matrix factorization literature. The norm of the $U$ and $V$ components are constrained and $D$ is diagonal. Experiments demonstrate that the UDV parameterization exhibits an implicit bias towards low rank for a wider range of learning rates and initialization sizes compared to the classical Burer-Monteiro factorization (BM; $X = UV^\top$, without norm constraints). Furthermore, the utility of the UDV factorization is evaluated as a replacement for linear layers in existing non-linear neural network architectures. The corresponding layers in the modified neural networks are shown to be low rank after training, enabling low rank pruning.

**Questions:**

--

**Ethical Concerns:**

["NO or VERY MINOR ethics concerns only"]

**Final Justification:**

As stated in the review and follow up comment to the authors' response, my assessment of the work is positive due to its clarity and the potential interest that the UDV factorization may attract, both from theorists and perhaps practitioners. However, due to the lack of a clear demonstration for the significance of UDV factorization, the recommended score is borderline accept. I will note that, if it were possible, I would have raised the overall score to a slightly higher one than borderline accept, somewhere in between borderline accept and accept (e.g., weak accept in similar conferences).

**Limitations:**

Yes

**Paper Formatting Concerns:**

--

**Quality:**

3

**Strengths And Weaknesses:**

Strengths
---
1. The paper is well-written and generally easy to follow. The experimental settings are described in sufficient detail, and the background motivating the design of the UDV factorization is clearly presented.

2. While a formal analysis of the UDV (or UDU variant) is not provided, the theoretical insight section, along with the accompanying experiments in Appendix A.3 are useful for comprehending why a low rank bias is observed.

3. The implicit low rank bias of BM factorization, or its deep variants where multiple unconstrained linear layers are multiplied, often heavily relies on a small initialization size. This creates a tradeoff between the low rank bias and convergence speed, since when the initialization is small the gradient (and higher-order derivatives for deeper factorizations) vanish. I find it interesting that the low rank bias in the UDV factorization does not require a small initialization, and so is not at odds with convergence speed.


Weaknesses
---
1. As the paper does not include a formal treatment of the UDV factorization, the significance of its contributions relies primarily on empirical validation. However, I find the evaluation falls below standard for empirical works in the field. While there are quite a few experiments included, the considered datasets are either non-standard tabular datasets or MNIST. It is not clear why the authors chose specifically these datasets and did not consider more relevant settings. Even if compute is a major restriction (which does not seem to be the case since experiments were run on a A100 GPU according to Section 3.1.1), surely going beyond MNIST for classification should be possible.

2. Furthermore, the neural network results do not show that the UDV factorization brings noticeable improvements in terms of resulting performance (Table 1). The low rank bias can still be considered as a positive due to its potential use for pruning. But, while the idea makes sense, its evaluation is again limited to simplistic settings. There is also no comparison to alternative neural network pruning and compression techniques.

To be clear, the critique above is not regarding the quantity of experiments (there are plenty, including the appendix), but rather on their quality — how decisive are they and do they substantiate the significance of the proposed UDV factorization. The paper lacks convincing evidence of the factorization being useful in practice. I believe it would have been much better off if many of the existing experiments (especially those in the appendix) would have been replaced with a few convincing experiments in more realistic settings.

Overall, despite the limitations above, my assessment of the work is positive due to its clarity and the potential interest that the UDV factorization may attract, both from theorists and perhaps practitioners. Unfortunately, it is difficult for me to wholeheartedly recommend acceptance of a paper whose main contributions rely on experiments in such simplistic settings. Thus, the borderline accept rating.

More minor comments:
---
1. In Figure 1, the convergence rate of the UDU factorization is not impacted by the change in initialization scale. However, looking at the gradient updates (below Equation 5), they should shrink the smaller the initialization scale is. Why then is the convergence rate not affected? Is it because the initialization scale is not taken to be small enough?

2. In lines 42-43, it is stated that the key insight behind this work is that the implicit bias in the classical BM factorization is driven by divergent dynamical behavior. However, the paper does not contain any evidence supporting this claim. Rather, it focuses on the newly proposed factorization. I strongly recommend rephrasing this sentence as it is misleading and can confuse readers.

3. Similar to the comment above, I find the remark in the subsequent line 44 to be unclear and not substantiated in the paper. Why would “standard (convergent) dynamics” be associated necessarily just with data fitting?

4. Typo in line 107: “here matrix completion” should probably be “here is matrix completion”.

5. In Equation 6, what does it mean for $\phi (x)$ to be approximately equal to $y$? It is not clear where this $y$ comes from or what is the purpose of this notation.

---

> ### Author Rebuttal · Authors · 2025-07-30
>
> Thank you very much for your thoughtful and constructive evaluation of our work. We appreciate the time and care you dedicated to providing detailed feedback.
>
> We understand the reviewer’s critique regarding the experiment setup, but we would like to clarify that the primary goal of our neural network experiments is to demonstrate that the hypothesized behavior emerges consistently across a wide range of settings. Specifically, we designed these experiments to test the robustness of our observations across different problem setups, which include different optimization algorithms, different parameters (such as learning rate), different network architectures and tasks (regression, classification, transfer learning, fine tuning), different depth and activation functions, etc. The consistent emergence of structure across these diverse setups supports our claim that the proposed $UDV^\top$ reformulation (with constraints) of dense layers promotes a meaningful structural bias. We will revise the manuscript to clarify this intent and better distinguish the goal of these experiments from standard performance benchmarking.
>
> While we agree that exploring the scalability of the proposed factorization in large-scale benchmarks is a natural and important direction, we note that a current limitation is related to the slow early-stage convergence behavior of the $UDV^\top$ model. This pattern is also evident in our matrix factorization experiments, for instance, although $UDU^\top$ converges faster to a high-accuracy solution with $10^{-9}$ residual than the classical BM factorization, it takes longer to reach a moderate-accuracy threshold like $10^{-3}$. This behavior presents a challenge under constrained computational resources; while we have access to capable GPUs, restrictions such as limited session duration and the number of simultaneously available devices limit our ability to conduct extensive large-scale benchmarks. Nevertheless, we can provide some moderately scaled use cases to illustrate practical viability.
>
> That said, this work also serves as the foundation for an ongoing, more comprehensive project on the practical implementation of UDV-based formulations. We are currently exploring ways to improve early-stage convergence without sacrificing the structural bias induced by the factorization. The slow initial progress is linked to transition delays in learning successive columns. Increasing the depth of the diagonal layer (e.g., using a $UDD^\top U^\top$ formulation) appears to shorten these transition phases (tested so far only on matrix factorization). Likewise, incorporating adaptive preconditioning techniques or using signSGD-type methods has shown promise in addressing this concern. In parallel, we are developing a more comprehensive pruning framework based on UDV, for regression, classification, and language modeling tasks. This includes pruning strategies both during and after training. Preliminary results are encouraging: for example, using energy-based thresholding as an adaptive pruning criterion during training, we can prune 96.64% of the parameters in the UDV-factorized linear layers of a ViT model (trained on CIFAR-10) without degrading its generalization performance.
>
> We would also like to emphasize that the emergence of structural bias (such as low-rank tendencies, orthogonality and column sparsity) is not only relevant for practical purposes like pruning, but also valuable to understand in its own right. Even if approximately low-rank structure is sufficient for many applications, uncovering the mechanisms that drive low-rank behavior can provide deeper insight into the inductive biases of neural networks. In this context, the observation of a truly low-rank (rather than an approximate low-rank) representation can make the underlying dynamics more interpretable. In turn, understanding these phenomena may enable more principled architectures and optimization strategies in the future. The goal of this work is therefore to highlight structure-promoting behavior of the $UDV^\top$ reformulation, with the hope that future work can further explain and build upon these dynamics.
>
> Please find below our responses to the specific questions raised:
>
> **Q1:** This is a great question. We did experiment with very small initializations. Recall that we initialize $D$ as the identity matrix and control the scale via $\xi = ||U||_F$. For instance, with $\xi = 10^{-4}$, the initial matrix $X = UDU^\top$ becomes norm bounded by $||X||_F \leq ||U||_F^2 = 10^{-8}$. In fact, $\xi$ influences early convergence behavior, however this is not immediately visible in Figure 1 due to the wide y-axis range (from $10^{-13}$ to $10^1$). If you zoom in (unfortunately we are not allowed to upload a figure in this year's rebuttal, but you can zoom in the PDF viewer), the influence becomes clearer. This influence is more noticable in the noisy setting in the appendix, so please refer to Figure SM1, where with $\xi = 10^{0}$ the first column is learned as early as iteration 100–200, whereas it takes around 1000–2000 iterations with $\xi = 10^{-4}$. However, interestingly, we observe that larger $\xi$ values tend to introduce longer transition delays before the learning of subsequent components kicks in. This ultimately dissipates the initial advantage of having a large $\xi$, and the asymptotic convergence rates appear quite similar across different $\xi$.
>
> **Q2:** We understand the reviewer’s concern. Our framework is inspired by this hypothesis, as we introduce constraints to prevent divergence of $U$, and the theoretical insights section suggests that the structure in $U$ emerges as it attempts to grow in norm. That said, we will soften the statement and clarify that this is a motivating hypothesis rather than a claim we rigorously prove or analyze.
>
> **Q3:** This point is closely related to the previous question. Our intention was to contrast our perspective with the common approach in the literature. We will revise the wording to clarify that this is an informal insight that guided the development of our formulation, rather than a formal claim or strict dichotomy.
>
> **Q4:** Thank you for your careful reading. We will correct the typo.
>
> **Q5:** The symbol "$\approx$" is commonly used and widely accepted in the context of supervised learning  to denote the goal of fitting a function $\phi(\mathbf{x})$ to the observed outputs $\mathbf{y}$, prior to introducing a loss function that formalizes this approximation. At this point, we have not yet defined the training objective, hence we use "$\approx$" to suggest that the model aims to approximate the target outputs. The sentence preceding equation (6) clarifies that $(\mathbf{x}_i, \mathbf{y}_i) \in \mathbb{R}^d \times \mathbb{R}^c$ are data-points, and equation (6) further clarifies that $\mathbf{x}$ refers to input features and $\mathbf{y}$ to the corresponding target outputs (i.e., responses or labels).

---

> > ### Comment · Reviewer_g4tS · 2025-08-03
> >
> > Thank your for the detailed response and answering the questions in my review. I have read the response and other reviews carefully. I acknowledge that the main purpose of the paper is to demonstrate that a new type of matrix factorization yields a robust low rank implicit bias. Yet, due to the lack of a clear demonstration for the significance of this contribution, I believe that my initial weakly positive assessment of the paper is fair. I will note that, if it were possible, I would have raised the overall rating to a slightly higher one than borderline accept, somewhere in between borderline accept and accept (e.g., weak accept in similar conferences).

---

> > > ### Author Response · Authors · 2025-08-07
> > >
> > > Thank you very much for your thoughtful comments and feedback. We really appreciate the time and effort you dedicated to reviewing our work.
> > >
> > > We also wanted to mention that we have provided additional numerical results demonstrating the practical potential of UDV for pruning Transformer models. If of interest, the details of this experiment can be found in our recent response to Reviewer iVqn.

---

### Official Review · Reviewer_eCQr · 2025-06-19

**Clarity:** 3
**Significance:** 3
**Originality:** 3
**Rating:** 5
**Confidence:** 4

**Summary:**

This paper tackles a matrix completion problem solved with projected gradient descent and extends the well-known BM factorization in the algorithm. Orthogonality constraints are replaced with Frobenius norm-constraints and a positive diagonal matrix that multiplies between the matrices (like SVD) is introduced. Extensive experiments along with fixed point analysis show that this extension results in low-rank solutions for a wide range of hyperparameters investigated in the literature. When used for training fully-connected layers of deep neural networks, this approach also facilitates pruning while not significantly impacting the prediction performance.

**Questions:**

Are validation and test sets used interchangably (Table 1 vs. Section 3.1.2) or are they separate sets? they should be separate since it sounds like learning rate etc. are optimized over the validation set.

How is the singular value threshold for pruning selected in practice?

**Ethical Concerns:**

["NO or VERY MINOR ethics concerns only"]

**Final Justification:**

The authors addressed all of my comments

**Limitations:**

limitations on theoretical analysis including convergence are noted in the Conclusion section

**Quality:**

3

**Strengths And Weaknesses:**

The factorization approach for matrix completion is unique. Extensive experiments including variants, comparisons to weight decay and applications in computer vision and NLP frameworks strengthen the results. The paper is clearly written.

---

> ### Author Rebuttal · Authors · 2025-07-30
>
> We appreciate the time and effort you dedicated to reviewing our manuscript and your positive evaluation of our work. Below, we address the questions you raised:
>
> **Q1:** We thank the reviewer for pointing this out and appreciate the opportunity to clarify. We used only two data splits: one for training and one for evaluation. Throughout the manuscript, we referred to the evaluation set as the validation set, but this corresponds to the test set in the standard train/validation/test terminology. That is, the same evaluation split is referred to in both Table 1 and Section 3.1.2, and the metrics reported under “validation” and “test” are identical. Importantly, this split remained strictly unseen during training and was only used for performance reporting. We will revise the manuscript to use consistent terminology and eliminate confusion.
>
> Regarding hyperparameter selection: we *did not* tune hyperparameters based on this evaluation split. All methods were run using a fixed grid of standard hyperparameters (e.g., learning rates  1e-6, 1e-5, 1e-4, ...), and we report the best performance for each method from this grid. This is a common and practical approach for comparing methods under equal tuning budgets. Since no iterative tuning or model selection was performed using the evaluation set, its role aligns more closely with that of a test set. We will make this explicit in the revised manuscript.
>
> Additionally, please note that Table 1 shows a summary of our results, and the complete results (with all parameter choices) are reported in Tables SM2–SM5 in the supplementary material.
>
> **Q2:** For simplicity, throughout this response, "number of neurons" refers to the number of neurons in the diagonal ($D$) layer.
>
> - **For the regression tasks (HPART and NYCTTD):** The diagonal layer was already small, so we pruned one neuron at a time, gradually reducing the number of neurons while monitoring pruning performance. Results are shown in Figure 4 (main text) and Figure SM13 (appendix).
>
> - **For the image classification task (MNIST):** The classifier layer was wider. We applied a recursive proportional rule: at each step, we retained the top 90% of neurons (by singular value), evaluated performance, and repeated the procedure until only one neuron remained (see Figure SM13).
>
> On a closing note, we would like to highlight that the pruning experiments included in this paper serve as an initial proof of concept. This work lays the foundation for an ongoing, more comprehensive project focused on UDV-based pruning strategies across regression, classification, and language modeling tasks. In this follow-up in progress, we are exploring adaptive and flexible pruning strategies, supported by careful sensitivity analyses.

---

### Official Review · Reviewer_gj1s · 2025-06-29

**Clarity:** 4
**Significance:** 3
**Originality:** 2
**Rating:** 4
**Confidence:** 4

**Summary:**

This paper introduces a novel matrix factorization UDV that utilizes norm constraints on the outer factors and a diagonal term in the middle. The authors demonstrate through numerical simulation on synthetic PSD matrix completion that solutions found through this factorization are exactly low-rank compared to BM factorization. This factorization is evaluated for linear regression and replacing FC layers in image classifiers, where it results in compressed low-rank representations.

**Questions:**

+ From my understanding of the analysis section, the columns of U for which the corresponding lambda is nonzero are forced to be zero eigenvectors of the gradient, but this doesn't necessarily explain why this forces low-rank solutions. Are you assuming that the gradient is full-rank?
+ My intuition for why columns of U go to zero is that with the matrix norm constraint, it is better to increase the magnitude of columns of U aligned with the top subspace of the gradient (and shrink the other ones), but then this would just result in a rank-one matrix? Overall, if the analysis could be made more rigorous and distinguish the behavior of this factorization (as opposed to BM or weight decay) this would help with appreciating the proposed factorization.
+ I feel that rather than being a new neural network, this factorization can help compress dense matrices in modern architectures (e.g., transformers). Have the authors explored this direction, and if so, what is the reason it was not included?

**Ethical Concerns:**

["NO or VERY MINOR ethics concerns only"]

**Final Justification:**

My main issues with the paper are analysis of low-rank bias and experimental evaluation. I believe the authors' response has improved some clarity of the analysis, but it remains somewhat opaque and handwavy. On the other hand, the authors claim they will add experiments that would greatly improve the practical impact of the paper. Overall, I lean towards acceptance since I believe the paper is well-written and proposes an interesting idea to the community, but needs further analysis and evaluation in the future.

**Limitations:**

Yes.

**Quality:**

2

**Strengths And Weaknesses:**

The numerical simulations are convincing of the low-rank bias of the proposed factorization, although the practical implications are difficult to assess as the experiments are conducted on toy datasets. The theoretical insights on page 5 are greatly appreciated, but I was left wanting a more convincing explanation of why low-rank bias arises (see questions below). A major strength of the paper is its clear and detailed exposition. Even though the evaluations are quite limited, the proposed method seems quite promising for more classical directions (low-rank recovery) as well as more modern ones (compressing DNNs), and to my knowledge is novel. The current drawbacks of the paper IMO are the analysis and lack of more modern evaluations. If these could be addressed, it would help increase my score.

---

> ### Author Rebuttal · Authors · 2025-07-31
>
> We sincerely thank the reviewer for their careful reading of our manuscript and for the constructive feedback. Below are the responses for the questions raised:
>
> **Q1/Q2:** Thank you for this question and the opportunity to clarify this point. We would like to emphasize that the analysis in Section 2.3 is intended as a theory-driven insight, not a full analytical characterization. (We have also noticed some typos in this section, which will be corrected in the revised manuscript.) There are a few key points worth highlighting:
>
> - The fixed-point analysis reveals that there are no fixed points satisfying $||U||_F > \alpha$. This is a fundamental observation because it implies that the set of fixed points for the $U D U^\top$ formulation is equivalent to that of the classical BM factorization, under a simple change of variables (i.e., $U\_{BM} = UD^{1/2}$). Thus, the introduction of the diagonal matrix $D$ along with the constraints does not restrict the fixed-point set, and any structure that emerges is due to the optimization dynamics (rather than the parameterization alone.) However, we have observed a consistent low-rank bias across multiple optimization algorithms (in our neural network experiments). This suggests that the phenomenon is linked to shared features of the optimization process, rather than a specific solver or parameter choice. Hence, fixed-point analysis can be a useful tool for probing the underlying mechanism driving this behavior.
> - It appears that the observed structural bias is closely tied to the regime where the norm of $U$ tends to grow and the Frobenius norm constraint becomes active. In fact, we experimented with artificially large $\alpha$ values combined with small step sizes, which allowed convergence to occur without triggering the norm constraint. In such cases, the resulting solutions exhibit no additional implicit bias beyond what is already present in the classical BM factorization. This supports our view that the enhanced bias observed in the $UDU^\top$ formulation is directly connected to the projection step onto the Frobenius norm ball.
> - It is also worth emphasizing that we observe more than just a low-rank bias. The columns of $U$ tend to align along orthogonal directions, and $U$ exhibits column sparsity, with many columns converging to zero. Interestingly, even the columns that eventually vanish tend to remain orthogonal during training. These phenomena appear only when the Frobenius norm constraint on $U$ is triggered.
> - A closer examination of the fixed-point conditions, assuming that the Frobenius norm constraint is active, offers partial insight into the structural behaviors we observe. In particular, under this assumption, the analysis suggests that the columns of $U$ should either align with the negative eigenvectors of $\nabla f(X)$ or become zero. When combined with random initialization of $U$, this may help explain the empirical tendency of $U$’s columns to align along orthogonal directions. [Importantly, as discussed above, such fixed points do not actually exist: at any true fixed point, the Frobenius norm constraint is inactive. Nevertheless, one can argue that the intermediate steps where the projection becomes active influence the optimization trajectory. These projection steps may push the iterates toward directions that reflect the (hypothetical) solution of the constrained fixed-point system, even if the constraint is ultimately inactive at convergence.]
> - Why do we observe low-rankness? In fact, low-rankness emerges as a byproduct of column sparsity in $U$. But how does this sparsity arise? Looking at the update rules, applied columnwise to $U$ and entrywise to $D$, we have:
> $$\bar{u}_j \gets u_j - 2\eta \lambda_j \nabla f(X) u_j \quad \text{and} \quad \bar{\lambda}_j \gets \lambda_j - \eta u_j^\top \nabla f(X) u_j.$$
> Interestingly, both $\bar{u}_j$ and $\bar{\lambda}_j$ grow rapidly along directions aligned with the negative eigenvectors of $\nabla f(X)$, particularly the dominant ones. Suppose $\nabla f(X) u_j = -\beta_j u_j$ for some $\beta_j > 0$. Then the updates simplify to:
> $$\bar{u}_j \gets (1+2\eta\beta_j\lambda_j)u_j \quad \text{and} \quad \bar{\lambda}_j \gets \lambda_j + \eta \beta_j ||u_j||^2.$$
> In this regime, $\bar{\lambda}_j$ remains positive by definition, so projection onto the nonnegative orthant for $D$ is ineffective. More importantly, under this dynamic, $\lambda_j$ will continuously increase, as there is no mechanism to decrease it.
> - Now, we look at the fixed-point condition for the $D$ update (see condition (d) in the paper): when $\lambda_j > 0$, it must satisfy $u_j^\top \nabla f(X) u_j = 0$.
> But under the previous assumption that $u_j$ aligns with a negative eigenvector (i.e., $\nabla f(X) u_j = -\beta_j u_j$), this condition reduces to:
> $-\beta_j ||u_j||^2 = 0 ~ \Rightarrow ~ ||u_j|| = 0.$
> This implies that, for such directions, the fixed point dynamics encourage $u_j$ to vanish *numerically* (hence sparsity). Considering the update scheme, however, this vanishing effect occurs through projection onto Frobenius norm ball, but this happens only if another column $u_j'$ exhibits a faster growth. So the growth one columns (or very few columns) will dominate.
> - But then, why don’t we observe a rank-1 solution? If the fixed-point conditions described earlier were the only driving mechanism, one might expect that only the dominant negative eigenvector survives, leading to a rank-1 solution. However, the situation is more nuanced. While the projection step can drive $||u_j||$ numerically toward zero for non-dominant columns, the same condition must be satisfied by all columns with $\lambda_j > 0$, including the dominant one. Crucially, there is no mechanism to shrink the dominant column. Since projection alone cannot enforce the fixed-point condition for the dominant column $u_j'$, the optimization trajectory should push $u_j'$ and $X$ in a way to evolve their alignment such that $u_j'$ falls onto the nullspace of $\nabla f(X)$, i.e., towards $\nabla f(X) u_j' = 0$.
> - Finally, as the previously dominant column, initially aligned with a negative eigenvector, transitions toward the nullspace, its growth slows down and eventually halts. At this stage, the algorithm enters a transition phase, where it appears to have nearly converged. However, despite being numerically small, some of the other columns continue to grow, eventually becoming the next dominant column.
> - We emphasize that this analysis is of course an oversimplified explanation. In particular, the updates to $u_j$ and $\lambda_j$ occur simultaneously with changes in $X$. Although the analysis presents these effects as if they occur in a sequential narrative, they are in fact all intertwined. Still, the matrix factorization experiments exhibit these phases, as can be observed in part from Figures SM4 to SM9 in the Appendix.
>
> **Q3:** We indeed plan to extend our approach to model compression across both vision and language tasks. Specifically, we aim to apply the UDV factorization to all (or subsets of) linear projections in multi-head attention (Q/K/V/O) and MLP blocks of Transformer-based models (e.g., ViT for vision and GPT-style models for language). Our goal is to integrate SVD-based pruning both adaptively during training and as a post-training procedure, depending on deployment or fine-tuning requirements. We even obtained some preliminary results after the submission of this paper. For example, on CIFAR-10, we trained a ViT model from scratch using UDV-factorized layers and were able to prune at least 96.64% of the trainable parameters across all UDV-factorized layers, without any observable loss in generalization performance compared to the original ViT.
>
> That said, one current limitation in large-scale models is the slower early-stage convergence of UDV. This pattern is also observed in our matrix factorization experiments: although UDV reaches high-accuracy solutions (e.g., $10^{-9}$ residual) faster than the classical BM factorization, it takes longer to reach a moderate accuracy threshold (e.g., $10^{-3}$). This becomes a challenge especially when dealing with large-scale problems with constrained computational resources. The slower early-stage convergence emerges from the long transitional phases in between learning different columns. To address this, we have ongoing work on improving the practical implementation of UDV. These efforts are directly inspired by the fixed-point insights discussed in the previous response.

---

> > ### Comment · Reviewer_gj1s · 2025-08-04
> >
> > Thank you to the authors for the thoughtful response. I think I understand the intuition a *little* bit better but still confused about how a solution of rank > 1 is found, maybe the authors can think of another way to present this. On the other hand, the model compression result would be an invaluable addition. Based on this, I will increase my score to borderline accept, as I believe this idea would be of interest to the community.

---

> > > ### Author Response · Authors · 2025-08-07
> > >
> > > Thank you very much for your thoughtful comments and feedback. We really appreciate the time and effort you dedicated to reviewing our work.
> > >
> > > We would like to clarify why the initially dominant column does not lead to a rank-1 solution. The main reason is that as $U$ and $D$ evolve, so does $X$, and hence $\nabla f(X)$. Without loss of generality, suppose the first dominant column is $u_1$, with all other columns numerically approaching zero. Here, $u_1$ is dominant in the sense that it grows faster than all other columns, i.e., $||\bar{u}_1 - u_1|| \geq ||\bar{u}_j - u_j||$ for all other $j$, so that after each iteration the other columns shrink while $u_1$ grows.
> > >
> > > However, as shown at the beginning of the fixed-point analysis, there is no fixed point satisfying $||\bar{U}||_F \geq \alpha$. This means the growth of $u_1$ must also stop (assuming convergence to a stationary point). This happens when eventually we get $\nabla f(X) u_1 = 0$.
> > >
> > > To illustrate, consider $f(X) = \tfrac{1}{2} ||X-Y||_F^2$, for which $\nabla f(X) = X - Y$. Suppose $X$ is initialized near zero, so the gradient is initially close to $-Y$; hence, suppose $U$ is aligned with the eigenvectors of $Y$. If $u_1$ is aligned with the dominant eigenvector and the other columns are negligible, then $\nabla f(X) \approx u_1 \lambda_1 u_1^\top - Y$.
> > > As $u_1$ and $\lambda_1$ evolve, the corresponding eigenvalue $\beta_1$ of $\nabla f(X)$ decreases. When $u_1 \lambda_1 u_1^\top$ fully matches the corresponding component in $Y$, we have $\beta_1 = 0$, i.e., $\nabla f(X) u_1 = 0$.
> > >
> > > At this point, the algorithm appears nearly converged: $u_1$ has stopped growing since $\beta_1 = 0$, and the other columns grow very slowly due to their small norms. However, these other columns still have some growth (which looks almost like a multiplicative growth)
> > > $$\bar{u}_j \gets (1 + 2 \eta \beta_j \lambda_j) u_j$$
> > > thus, even if initially tiny, the next dominant column eventually grows at a noticeable rate.
> > >
> > > Finally, we also wanted to mention that we have provided additional numerical results demonstrating the practical potential of UDV for pruning Transformer models. If of interest, the details of this experiment can be found in our recent response to Reviewer iVqn.

---

### Official Review · Reviewer_iVqn · 2025-07-02

**Clarity:** 3
**Significance:** 2
**Originality:** 2
**Rating:** 3
**Confidence:** 4

**Summary:**

This paper proposes a matrix factorization model, $UDV^\top$, where $U$ and $V$ are explicitly constrained to have a bounded Frobenius norm, and where D is a diagonal matrix with (explicitly enforced) nonnegative entries. The authors motivate this decomposition through the implicit bias of gradient descent to low-rank solutions, and report that it finds truly low-rank solutions, whereas a standard $UV$ factorization only finds approximately low-rank solutions. Based on these results, a new neural network architecture is proposed. Experiments show that low-rank solutions are obtained, resulting in a natural pruning strategy.

**Questions:**

1. Can you provide a motivation for considering the nonnegativity constraints on $D$? Do you have an explanation for why it works better than without nonnegativity constraints?
2. Are there other practical advantages to having an architecture that converges to low-rank solutions, besides for pruning?
3. Do you think gradient clipping, which entails a modification to the optimizer instead of the architecture itself, can be related to your approach?

**Ethical Concerns:**

["NO or VERY MINOR ethics concerns only"]

**Final Justification:**

For me, this work has two significant weaknesses: (1) the limited theoretical analysis regarding convergence to low-rank solutions, and (2) the unclear significance of the proposed factorization. During the discussion period, the authors have attempted to address the latter concern by (a) describing more explicitly the potential advantages of low-rank neural network models, and (b) conducting a more sophisticated pruning experiment. While I appreciate these efforts, it remains difficult to assess the merits of the proposed factorization. In this regard, a proper comparison with existing pruning techniques, or another convincing illustration of a practical benefit of the proposed factorization, is required, I believe.

To be clear, I agree that the proposed factorization is potentially interesting, but in my opinion the paper needs further strengthening, either through stronger theoretical results or through convincing experiments regarding the practical advantages.

**Limitations:**

yes

**Quality:**

2

**Strengths And Weaknesses:**

Strengths:
- The paper is in general well-written, and key ideas are presented in a simple manner.
- It is interesting that the proposed factorization has an implicit bias to low-rank solutions, even for large step sizes and large distances to the origin, and when used in neural network architectures.

Weaknesses:
- The theoretical motivation for, and analysis of the factorization are limited. The authors provide no theoretical guarantees for converging to low-rank solutions, although the `theoretical insights’ in section 2.3 provide some loose justification based on a fixed-point analysis. Moreover, the motivation for even considering nonnegative matrices $D$ is unclear.
- The significance and impact of the proposed architecture is not sufficiently motivated. The main advantage of the proposed architecture is that it converges to low-rank solutions. It should be better argued why this is desirable. The only motivation given in the paper is that this results in a natural pruning strategy, without compromising significant performance. However, the authors do not sufficiently substantiate this last claim, as they do not compare to any existing pruning technique.

Typos:
1. Line 107: A notable example here matrix completion -> A notable example here is matrix completion
2. Inconsistency: subj. to in eq. (1), s.t. in eq. (4)
3. Inconsistency: step-size versus step size. In Lines 247-248 you first use step sizes and then LRs
4. Inconsistency: Lines 671-672 write 1e-3 and 1e-4, whereas the rest of the paper uses 10^{-3} and 10^{-4}

---

> ### Author Rebuttal · Authors · 2025-07-30
>
> We would like to thank the reviewer for careful reading and for pointing out some typos. We will correct these in the revision.
>
> We would like to clarify that the primary goal of our neural network experiments is to demonstrate that the hypothesized behavior emerges consistently across a wide range of settings. Specifically, we designed these experiments to test the robustness of our observations across different problem setups, which include different optimization algorithms, different parameters (such as learning rate), different network architectures and tasks (regression, classification, transfer learning, fine tuning), different depth and activation functions, etc. The consistent emergence of structure across these diverse setups supports our claim that the proposed $UDV^\top$ reformulation (with constraints) of dense layers promotes a meaningful structural bias.
>
> The pruning experiments included in the paper serve as an initial proof of concept. This work forms the basis for an ongoing, more comprehensive project focused on UDV-based pruning strategies across regression, classification, and language modeling tasks, including pruning both during and after training. Preliminary results are encouraging: for instance, by applying energy-based thresholding during training, we are able to prune 96.64% of the parameters in the UDV-factorized linear layers of a ViT model trained on CIFAR-10 without compromising generalization performance.
>
> We address your questions below:
>
> **Q1:** The nonnegativity of $D$ arises naturally in our matrix factorization framework, where we focus on positive semidefinite (PSD) factorizations. A matrix of the form $U D U^\top$ is PSD if and only if the diagonal entries of $D$ are nonnegative. In the context of neural network training, this choice is largely a matter of convention rather than necessity. Importantly, the search space remains unchanged regardless of whether we impose nonnegativity on $D$, since there is no sign constraint on the entries of $U$ and $V$. For any matrix $U D V^\top$ with a negative entry in $D$, one can equivalently absorb the sign into the corresponding column of $U$ or row of $V$, resulting in an equivalent representation with a nonnegative $D$. We therefore adopted the nonnegativity convention in this case mainly to resemble a classical singular value decomposition, especially given our empirical observation that the columns of $U$ and $V$ tend to become orthogonal. For completeness, we also conducted experiments without the nonnegativity constraint on $D$ and reported them in the supplementary material (see Appendix B.1 and the corresponding figures and tables). The results were similar, although the UDV variant (main variant we used, with nonnegativity constraint on $D$) slightly outperforms the UDV-s (the variant without nonnegativity constraint on $D$) version in most cases; possibly due to the removal of redundant sign ambiguity.
>
> **Q2:** Yes, there are several practical advantages to having low-rank neural network models. Besides model compression, efficient pruning, and distillation [R1, R2], low-rank solutions enable model deployment on resource-constrained devices and efficient communication in distributed training scenarios [R3]. Empirical studies have demonstrated that low-rank structures can lead to improved generalization and reduced overfitting [R4, R5]. Additionally, they have been linked to increased robustness [R6, R7], enhanced interpretability of learned representations, and improved reasoning performance in large language models [R8], among other benefits.
>
> We would also like to emphasize that the emergence of structural bias (such as low-rank tendencies, orthogonality and column sparsity) is not only relevant for practical purposes like pruning, but also valuable to understand in its own right. Even if approximately low-rank structure is sufficient for many applications, uncovering the mechanisms that drive low-rank behavior can provide deeper insight into the inductive biases of neural networks. In this context, the observation of a truly low-rank (rather than an approximate low-rank) representation can make the underlying dynamics more interpretable. In turn, understanding these phenomena may enable more principled architectures and optimization strategies in the future. The goal of this work is therefore to highlight structure-promoting behavior of the $UDV^\top$ reformulation, with the hope that future work can further explain and build upon these dynamics.
>
> [R1] Denil et al., Predicting parameters in deep learning. 2014.
>
> [R2] Denton et al., Exploiting linear structure within convolutional networks for efficient evaluation, 2014
>
> [R3] Goyal et al., Compression of deep neural networks by combining pruning and low rank decomposition, 2019.
>
> [R4] Huh et al., The low-rank simplicity bias in deep networks, 2021
>
> [R5] Pinto et al., On generalization bounds for neural networks with low rank layers, 2024
>
> [R6] Savostianova et al., Robust low-rank training via approximate orthonormal constraints, 2023
>
> [R7] Langenberg et al., On the effect of low-rank weights on adversarial robustness of neural networks, 2021
>
> [R8] Sharma et al., The truth is in there: Improving reasoning in language models with layer-selective rank reduction, 2023
>
>
> **Q3:** This is an interesting question. However, we are not aware of any direct connection between gradient clipping and low-rank implicit bias, either theoretically or empirically. Gradient clipping modifies the optimization process by constraining update magnitudes, primarily to mitigate issues such as exploding gradients. In contrast, our formulation modifies the architecture itself, hence changes the optimization landscape in a structural way, which is fundamentally different than clipping. It might be interesting to apply gradient clipping within the $UDV$ formulation to investigate whether it has any impact (positive or negative) on the structural bias we observe.

---

> > ### Comment · Reviewer_iVqn · 2025-08-04
> >
> > Dear authors, thank you for the clarifications. I understand that the primary goal of this work is to show the emergence of a structural bias to low-rank solutions, and I acknowledge that this may have several practical advantages, as described in your answer. Yet, in my opinion, the current work still lacks either a thorough theoretical analysis or a practical demonstration of the advantages of the proposed factorization, and it appears that the other reviewers have raised similar concerns. While the proposed factorization is potentially interesting, I believe that the aforementioned limitations should be addressed to make the paper more convincing.

---

> > > ### Author Response · Authors · 2025-08-07
> > >
> > > Thank you for your comments and engagement during the discussion period.
> > > We present below an additional experiment on pruning for Transformer models to address your concern regarding the practical applicability of our proposed method.
> > > Let us first highlight our key finding: **UDV factorized ViT-based model trained on CIFAR-10 dataset by 95% in trainable parameters and 88% in FLOPs.
> > > The structured pruning is performed during training, which reduces both training and inference costs.**
> > > This significantly improves overall training and inference efficiency, demonstrating the practical significance of UDV for efficient real-world deployment of Transformer models.
> > >
> > > Please see the experiment details below:
> > >
> > > The baseline model architecture is based on the Vision Transformer (ViT) [1]. Since we are using the CIFAR-10 dataset, we use an input size of 32×32×3 and a patch size of 4×4 to accommodate the lower resolution.
> > > This model consists of 6 Transformer blocks, each with a hidden size (d_model) of 512 and 8 attention heads. The MLP hidden dimension within each block is also set to 512.
> > >
> > > We use the AdamW optimizer with an initial learning rate of 1e-4. To detect the plateau phase during training (i.e., when the training loss no longer decreases significantly), we employ an Exponential Moving Average (EMA)-based monitoring strategy. Once a plateau is detected, we apply a CosineAnnealingLR scheduler to gradually reduce the learning rate to 5% of its original value. This learning rate annealing helps to stabilize training and encourages the model to converge to a better local minimum.
> > >
> > > From a structural perspective, our proposed UDV method factorizes all linear layers in the Transformer blocks, including the query (Q), key (K), value (V), output (O) projections, as well as the MLP layers, except for the final classification layer.
> > > We denote the model used for pruning as UDV-ViT.
> > > The training setup, including random seed and all shared hyperparameters, is kept identical to the baseline model to ensure fair comparison.
> > > Both models are trained for 500 epochs with a batch size of 128.
> > >
> > > At initialization, UDV-ViT maintains approximately the same number of trainable parameters as the baseline model.
> > > We adopt the same Exponential Moving Average (EMA)-based monitoring strategy as the baseline to determine the epoch at which pruning should first be applied.
> > > Subsequent pruning steps are decided by a pruning scheduling strategy: if the performance drop after pruning is acceptable, the next pruning step proceeds immediately;
> > > otherwise, we wait for the performance to recover before continuing.
> > > To avoid long delays, we limit the maximum number of waiting epochs.
> > >
> > > Unlike the previous experiments presented in this paper that adopt Pruning After Training (PAT), where pruning is applied after a fully trained model is obtained, our approach follows the Pruning During Training (PDT) paradigm.
> > > PDT dynamically scales the model during training, which not only reduces inference cost but also decreases training computation and memory usage.
> > >
> > > We propose two PDT strategies:
> > > 1. Global pruning: This strategy performs pruning across all factorized layers simultaneously. It consists of four pruning rounds, with singular value energy thresholds set to [0.999, 0.997, 0.995, 0.99].
> > > 2. Block-wise pruning: This strategy involves two complete pruning rounds. In each round, we sequentially prune one Transformer block at a time, starting from the last block and proceeding to the first (i.e., from deep to shallow). Each pruning step applies to factorized layers within a single block and is controlled by the pruning scheduling strategy. After all blocks have been pruned once, a final global pruning step is applied. The singular value energy thresholds used for the two rounds are [0.999, 0.995], respectively.
> > >
> > > In both strategies, pruning decisions are based on the cumulative energy retained by the singular values of the factorized matrices.
> > >
> > > The results are summarized below:
> > >
> > > | Method | Parameters (millions) | FLOPs (giga) | Test accuracy (%) |
> > > |-|-|-|-|
> > > | Baseline | 9.51 | 1.28 | 72.8 |
> > > | UDV + global pruning | 0.42 | 0.15 | 72.6 |
> > > | UDV + block-wise pruning | 0.43 | 0.16 | 72.7 |
> > >
> > > An 88% reduction in FLOPs translates into a substantial decrease in inference time, while a 95% reduction in the number of parameters corresponds to significant savings in memory and storage,  without essentially compromising test accuracy. These results demonstrate the promise of UDV as a practical tool for deploying compact and efficient Transformer models. We will include these results in the final version of our paper, along with similar findings under varying parameter choices (e.g., batch size, step size, energy thresholds).
> > >
> > > [1] Dosovitskiy, et al. (2021). An Image is Worth 16x16 Words: Transformers for Image Recognition at Scale.

---

> > > > ### Comment · Reviewer_iVqn · 2025-08-08
> > > >
> > > > Dear authors, I appreciate your effort in providing additional pruning experiments, and I will increase my rating to 3. However, it remains difficult to assess how your results compare to existing pruning schemes, and therefore to judge the significance of the proposed factorization. Combined with the limited theoretical analysis, I believe this manuscript would benefit from further work to more convincingly present the merits of the proposed factorization.

---

> > > > > ### Author Response · Authors · 2025-08-09
> > > > >
> > > > > Thank you for your engagement in the discussion phase and for reconsidering your score.

---

### Note · Authors · 2025-08-16

Dear all,

Thank you for your valuable time and comments.
We take this opportunity to highlight our contributions and strengths.

Motivated by the insight of (Razin & Cohen, 2020) that low-rank implicit bias in matrix factorization (MF) and NN training persists for models that diverge (and even intensifies, as in their case the rank decreases towards its minimum), we hypothesize that such bias may be shaped by the forces driving divergence. To explore this, we propose a new factorization $UDV^T$ (inspired by the SVD structure) with Frobenius norm constraints on $U$ and $V$ to prevent their divergence and a diagonal component $D$ to retain the ability to span the full space.

Our extensive experiments on MF and NN training led to several critical insights we believe are valuable to the community:

* Compared to Burer–Monteiro (BM) factorization, the proposed model consistently demonstrates truly (rather than approximately) low-rank solutions regardless of step size and initialization, a property found to be **interesting**,  **appealing** (*iVqn / g4tS*) and **unique** (*eCQr*). The resulting factors also show a bias for column-sparse and orthogonal structures. These provide **convincing** empirical support (*gj1s*).
* A fixed-point analysis shows that the first-order stationary points of our formulation coincide with those of BM. The analysis also offers *partial* theoretical insight into the structural tendencies such as orthogonal alignments and column sparsity. These insights were **greatly appreciated** (*gj1s*).
* Applied to NN, the proposed model consistently exhibits a low-rank bias across a wide range of settings (varying in network depth, activations, optimizers, initialization, learning rates, architectures and tasks). This behavior persists even when compared against models with weight decay. These results were noted **interesting** and **promising** (*iVqn / gj1s*).
* By leveraging the low-rank solutions, we demonstrate significant NN compression via SVD-based pruning as a proof-of-concept; noted as a **novel** contribution (*gj1s*). To address concerns about practical applicability under modern architectures, we implemented a pruning-during-training method in a large transformer-based model during discussions; see our reply to (*iVqn*). The model compression results were recognized as an **invaluable addition** and are expected to be of **interest to the community** (*gj1s*).
* Finally, the work has been noted for its clarity by all reviewers.

---

### Decision · Program_Chairs · 2025-09-17

**Decision:**

Reject

**Comment:**

This paper introduces a novel matrix factorization, UDV, which imposes norm constraints on the outer factors and incorporates a diagonal term in the middle. Numerical simulations on synthetic PSD matrix completion demonstrate that this factorization yields exactly low-rank solutions, in contrast to BM factorization.

While the factorization approach is potentially interesting, two major weaknesses limit the contribution: (1) the paper provides only intuitive explanations, with limited theoretical analysis of convergence to low-rank solutions, and (2) the significance and practical impact of the proposed factorization remain unclear. The authors do not necessarily need to address both, but in the absence of stronger theoretical analysis, the current experiments appear somwwhat insufficient to support claims of practical utility. The paper would benefit from either more rigorous theoretical guarantees or more convincing experimental evidence demonstrating practical advantages.